Remote sensing pipeline for tree segmentation and classification in a mixed softwood and hardwood system

McMahon Conor A. conor.mcmahon@utexas.edu
Department of Mechanical Engineering, University of Texas at Austin , Austin, TX , USA
Boyer Alison
Electronic publication date: 2019 Feb 28
Publication date: 2019
Volume: 6
Electronic Location ID: e5837
Received 2018 Jun 4; Accepted 2018 Sep 27
Copyright: © 2019 McMahon
Copyright year: 2019
Copyright holder: McMahon
License: This is an open access article distributed under the terms of the Creative Commons Attribution License, which permits unrestricted use, distribution, reproduction and adaptation in any medium and for any purpose provided that it is properly attributed. For attribution, the original author(s), title, publication source (PeerJ) and either DOI or URL of the article must be cited.
License URL: https://creativecommons.org/licenses/by/4.0/

Keywords: Remote sensing, Forestry, Lidar, Hyperspectral camera, Segmentation, Classification, Alignment, Ecology, Data science, Biogeography

Funding: The National Science Foundation Battelle Memorial Institute The National Science Foundation through the NEON Program NIST IAD Data Science Research Program The Gordon and Betty Moore Foundation’s Data-Driven Discovery Initiative GBMF4563 An NSF Dimension of Biodiversity program DEB-1442280 The National Ecological Observatory Network is a program sponsored by the National Science Foundation and operated under cooperative agreement by Battelle Memorial Institute. This material is based in part upon work supported by the National Science Foundation through the NEON Program. The ECODSE competition was supported, in part, by a research grant from NIST IAD Data Science Research Program to D. Z. Wang, E. P. White, and S. Bohlman, by the Gordon and Betty Moore Foundation’s Data-Driven Discovery Initiative through grant GBMF4563 to E. P. White, and by an NSF Dimension of Biodiversity program grant (DEB-1442280) to S. Bohlman. These funding sources allowed the collection of the data used in the competition and the specification of the competition rules. The authors received no resources from these organizations outside of the data provided for the competition. There was no additional internal or external funding received for this study. The data provided for the competition were provided by the National Ecological Observatory Network as described in the Funding Statement. Provision of this data was the only manner in which resources were provided by that organization. The funders had no role in study design, data collection and analysis, decision to publish, or preparation of the manuscript.

==============================
The National Institute of Standards and Technology data science evaluation plant identification challenge is a new periodic competition focused on improving and generalizing remote sensing processing methods for forest landscapes. I created a pipeline to perform three remote sensing tasks. First, a marker-controlled watershed segmentation thresholded by vegetation index and height was performed to identify individual tree crowns within the canopy height model. Second, remote sensing data for segmented crowns was aligned with ground measurements by choosing the set of pairings which minimized error in position and in crown area as predicted by stem height. Third, species classification was performed by reducing the dataset’s dimensionality through principle component analysis and then constructing a set of maximum likelihood classifiers to estimate species likelihoods for each tree. Of the three algorithms, the classification routine exhibited the strongest relative performance, with the segmentation algorithm performing the least well.

Background

Characterizing the structure and species makeup of forest systems is an important task in many disciplines. These kinds of analysis are necessary for assessing the quality of a patch of habitat for conservation of particular target taxa (Rose et al., 2015; Fletcher & Erskine, 2012), for estimating system-level properties like primary productivity or capacity for carbon sequestration (Vassallo et al., 2013), and also for landowners interested in directly managing forests for wood or fruit production (Tang & Shao, 2015). Traditional methods of characterizing forests involved expensive, laborious, and time-consuming deployment of experts on foot to manually label individual trees with location, species, and structural data (Barbosa & Asner, 2017; Marconi et al., 2018). More recently, remote sensing technologies have emerged which show the potential to massively alter the scale and efficacy with which these characterizations can be performed. These technologies utilize aircraft (small planes or unmanned aerial vehicles, or in some cases satellites) to fly cameras and light detection and ranging (LiDAR) units over forests, collecting massive amounts of data on the structural and spectral properties of the communities (Tang & Shao, 2015; Mulla, 2013).

Extracting useful ecosystem parameters from this mass of generated data involves three primary steps: segmentation, alignment, and classification (Marconi et al., 2018). In the segmentation step, individual tree crowns (ITCs) are automatically extracted from the scene so that they can be counted and analyzed separately. During alignment, individual trees from the segmented scene are automatically paired with stems labeled during traditional ground-based methods to improve the richness of the remote sensing dataset. This also allows assignment of species labels to some crowns, which provides training data for the classification step. During classification, species labels are estimated for remaining trees which were not already assigned labels by experts on foot.

In general the efficacy of different remote sensing processes depends strongly on the forest type being surveyed—in particular the degree of canopy openness and overall species diversity (Naidoo et al., 2012; Zhen, Quackenbush & Zhang, 2016). When new methods are introduced in the literature there is often a lack of robust comparison to existing methods, and the comparisons which are included are difficult to apply broadly due to these inherent differences in performance on different systems (Marconi et al., 2018; Zhen, Quackenbush & Zhang, 2016). As well, the formats in which remote sensing data are saved and processed vary hugely across platforms and research disciplines, and have proven difficult to standardize (Marconi et al., 2018; Zhen, Quackenbush & Zhang, 2016).

The data science competition

The National Institute of Standards and Technology (NIST) data science evaluation (DSE) plant identification challenge (Marconi et al., 2018) was introduced this year to try to combat these problems and to increase standardization, methods benchmarking, and collaboration within the remote sensing research community. During this competition, the three tasks outlined above (segmentation, alignment, and classification) were performed on the same dataset by multiple competing teams. To keep the results of each task from limiting the performance of subsequent tasks, the input data for each task was provided a priori by the competition organizers. In a real-world scenario the tasks could instead be arrayed in a single coherent pipeline to perform more meaningful automatic forest characterization. More detailed descriptions of the data will be provided in the method subsections following, and exact methods of input data collection and preprocessing can be found in the parent paper on the overall competition pilot, along with elaboration on the nature and goals of the competition (Marconi et al., 2018).

The input data used for the competition were collected by National Ecological Observatory Network (NEON) over the Ordway-Swisher Biological Station (Domain D03, OSBS) from 2014 to 2017 (Marconi et al., 2018). The field site has a mix of species, with the two most common being the coniferous longleaf pine (Pinus palustris) and the deciduous turkey oak (Quercus laevis). The forest is largely split into three distinct ecosystem types: evergreen forest, emergent herbaceous wetland, and woody wetland (Marconi et al., 2018). This mixed softwood and hardwood system provides a suitably complex combination of different habitats and canopy structures to ensure that segmentation and classification efforts are challenged adequately.

Following is a discussion of prior work within the segmentation and classification subtasks, both of which represent well-studied areas in the remote sensing field.

Segmentation

Many segmentation algorithms have been investigated by remote sensing researchers, with results that vary strongly based on algorithm choice (Kaartinen et al., 2012) and on system structural parameters, with more structurally complicated systems generally being more difficult to segment (Naidoo et al., 2012). LiDAR sensors produce point clouds made up of unordered XYZ points, and some segmentation algorithms are run on these point clouds. However, many others are instead run on a raster gridded canopy height model (CHM) which is created via postprocessing of the point cloud into a more convenient matrix format, with each pixel in the grid containing the local height of the canopy above the ground (Van Leeuwen & Nieuwenhuis, 2010). CHM models are more simple, less storage-intensive, and may be less computationally expensive than point cloud models. This comes with the disadvantage that CHM models contain less information than point clouds, and understory trees are often completely omitted, although some workers have attempted to ameliorate this problem (Van Leeuwen & Nieuwenhuis, 2010; Lee & Lucas, 2007).

One common CHM method involves labelling of local maxima in the image (representing the peaks of ITCs) and subsequent watershed segmentation about those peaks into discrete trees (Van Leeuwen & Nieuwenhuis, 2010). One point of variability within these marker-controlled watershed systems is the means of selection of local maxima. The primary issue here is that a single tree crown may actually include multiple maxima if it is not perfectly conical or spherical—for example, a tree may have several separate branching sections which each contain maxima. Careless selection of local maxima may result in oversegmentation of single trees into multiple modeled crowns. One method to deal with this is the selection of a varying window size in which to look for local maxima. Popescu & Wynne (2004) put forward a method in which window sizes are given by a linear function of tree height. This method is effective in cases where tree height is strongly correlated with crown diameter, because then the window can be effectively set to the diameter of the target tree based on the height of each pixel being considered, allowing exclusion of maxima from other trees from consideration while locating the maximum of a given tree.

An approach which attempts to correct for shortcomings in CHM approaches was presented by Van Leeuwen, Coops & Wulder (2010). The authors used a modified Hough transform to search for conical shapes directly in the point cloud space, using a defined range of cone angles and heights. Their algorithm allows sharp delineation of borders between trees (intersections of cones) and precise specification of tree tops (cone maxima, or points). It also creates a much less data-intensive cloud, as each cone can be represented with only a few parameters, eliminating the need to store every pixel on the cone’s surface within the CHM and consequently reducing the data size by 80 times. The resulting parametric height model closely matched with both the CHM and the actual ground-truth tree data. However, the authors note that their study system may be unusually suited to this kind of analysis, because it consisted of a plantation of coniferous trees, all of which were relatively young (mostly free from broken branches) and of similar shape and structure (conical). In more complicated forests featuring many different crown shape types, such as the OSBS, this kind of method may not be feasible.

Another approach which works directly on point clouds has been investigated by Morsdorf et al. (2004). The authors first found local maxima from the pixel-based digital surface model (the digital surface model is analogous to the CHM but gives actual canopy height, not the height above the ground). Next, k-means clustering was used in XYZ to create masses of points (representing trees) around each maximum. The clustering was based on simple Euclidean distances, with Z being weighted differently from X or Y to account for the vertical ellipsoid shape of tree canopies in the boreal, coniferous system in which the study was conducted. The work was successful at predicting the heights of trees that were detected (R2 = 0.923) but was much less successful at estimating crown area (R2 = 0.204). The authors believe this is because of the algorithm’s tendency to clump trees which are nearby one another into single tree masses—this tendency is also reflected in their relatively high omission error (finding 1,200 trees in a 1984-tree stand). This shortcoming may make their approach unattractive for systems with low canopy openness in which trees are packed tightly against one another. As well, this kind of algorithm is again suited best for cases where tree shape is reasonably predictable (vertical ellipsoids).

Comparisons between different segmentation methods have been difficult because of the high dependency of success on system characteristics like tree shape, canopy openness, variation in tree age, species composition, and other factors (Marconi et al., 2018; Kaartinen et al., 2012). Additionally, the actual point density produced by the LiDAR within the output cloud may vary substantially across studies, with denser clouds generally allowing better discrimination between individual crowns (Zhen, Quackenbush & Zhang, 2016). One study expressly sought to address these problems by comparing many different segmentation techniques on a single common data set (Kaartinen et al., 2012). A number of different techniques were used and their results were evaluated in terms of the number of correctly matched trees, the number of missed trees, the accuracy in XY location and height of the identified trees, and comparisons of the predicted to the expected crown area. The best algorithm investigated in their study was based on the minimum curvature within the CHM, with each CHM point being scaled by its local minimum curvature prior to local maxima finding and marker-controlled watershed segmentation.

Due to its relative simplicity and the existence of open source software to support its application, marker-controlled watershed segmentation with a variable-sized maxima search window was selected for the segmentation task in this competition. This algorithm is similar to (but simpler than) the best-performing algorithm discussed in the comparative paper above (Kaartinen et al., 2012). The authors of that study also tested an algorithm very similar to this one and it performed only slightly less well in most metrics than the best algorithm used (see references to the method FGI_VWS in that work).

Classification

One old and well-studied method for species classification is Gaussian maximum likelihood. This method represents each tree class using a signature template given by a list of normal distributions in each feature variable to be used during classification. Templates describe the distribution of expected values in each variable for the species class in question. Each target tree is decomposed into a test vector in feature space and compared to every template, and the likelihood of the test vector belonging to each template distribution is calculated. The most likely species class may then be assigned as the tree’s species label. Maximum likelihood methods have achieved high classification accuracy in some test systems (Sisodia, Tiwari & Kumar, 2014; Dalponte et al., 2013; Clark, Roberts & Clark, 2005), but may perform poorly when the feature space is highly dimensional and the training data for classes are unevenly sampled, causing the covariance matrices necessary for the likelihood model formulation to be estimated poorly (Naidoo et al., 2012). Nonetheless, for cases where it is able to perform, maximum likelihood is useful for its simplicity and its parametric nature, allowing the user to create actual models of the expected tree signatures in feature space.

Spectral angle mapping (SAM) is another common classification technique used in remote sensing (Clark, Roberts & Clark, 2005). This method also creates signature spectra of training trees in the feature space, but models them only as mean values, not distributions. Then, test trees are compared to the list of training trees and the training vector with the smallest angular deviation in feature space from the test vector is selected as the best species class. SAM approaches have been implemented which treat tree classes with either single or multiple signature spectra for each species (Cho et al., 2010). The latter approach can be leveraged to account for differences in plant phenology across individuals which result in multiple signature modes within the feature space for each class.

More complicated systems for classification include support vector machines (Dalponte et al., 2013; Heinzel & Koch, 2012), artificial neural networks (ANN), random forest systems (Dalponte et al., 2013), and object-based image analysis (Blaschke, 2010). Although these systems have shown promise in some applications, they require greater complexity in implementation and in some cases computation time (Naidoo et al., 2012), and for some systems (especially ANN (Blaschke, 2010)) may require substantial training data. Consequently, these systems were not considered for this project due to the time constraints of the competition and the small training sample size for some tree species. Instead, maximum likelihood was selected for the first round of competition tests, largely due to its relative simplicity of implementation and proven record in many test systems (Sisodia, Tiwari & Kumar, 2014; Dalponte et al., 2013; Clark, Roberts & Clark, 2005).

Methods

Data collection

All data used were provided by NEON (Keller et al., 2008; National Ecological Observatory Network, 2016), and included: Woody plant vegetation structure (NEON.DP1.10098)—hand-labeled data on tree species, height, diameter, location, and stem-crown correspondences;

Spectrometer orthorectified surface directional reflectance—flightline (NEON.DP1.30008)—hyperspectral data representing the canopy in 426 spectral bands;

Ecosystem structure (NEON.DP3.30015)—LiDAR-produced CHM;

High-resolution orthorectified camera imagery (NEON.DP1.30010)—RGB images of the scenes in question.

Again, more information regarding the data provided can be found in the parent paper (Marconi et al., 2018).

Segmentation

Input data

For this task, only the CHM and hyperspectral camera imagery were used. The dataset consisted of 30 training and 13 test plots, each a pair of CHM and hyperspectral camera images of the forest in an 80 × 80 m area, including a 20 m buffer on each edge of the 40 × 40 m plot. Both rasters were gridded with 1 × 1 m pixel sizes such that they contained 6,400 points each. The hyperspectral image contained 426 bands between 350 and 2,500 nm, while the CHM image contained only a single band filled with height values. More details on the methods of collection for the input data can be found in the parent paper (Marconi et al., 2018).

Processing overview

The approach used started with application of a normalized differential vegetation index (NDVI) filter to the CHM image in order to ignore pixels that had NDVI values too low to be plant matter. Next a variable-sized window to identify local maxima (presumptive treetops) was applied. Finally, a watershed segmentation to create tree polygons around these top points was performed.

Vegetation filtering

Normalized differential vegetation index is an index used to determine the degree of plant cover at a point in a spectral scene (Mulla, 2013). It is given as: (1) NDVI=NIR−REDNIR+RED

where NIR and RED are reflectances of the scene in the red and near infrared bands. For this filter bands 50 and 70 of the hyperspectral image were used, which correspond to wavelengths of 628.1 and 728.3 nm (respectively in the red and near-red IR ranges). High positive values of NDVI indicate substantial plant cover. Low or negative NDVI indicates land cover by non-vegetative materials. A threshold of 0.5 was used for filtering because this removed most of the ground cover from the image while maintaining all of the canopy material—this was confirmed via manual inspection of the RGB imagery pre- and post-filtering.

Maxima search

A search for local maxima (treetops) was performed on the NDVI-filtered CHM using the open source R package ForestTools (Plowright, 2018). The TreetopFinder() function within that package was used, which implements an algorithm presented by Popescu & Wynne (2004). This algorithm searches in a variable-diameter spatial window around each pixel in the image, and if the pixel’s height is the greatest within that window it is marked as a local maximum, or tree top. The window radius used was given by the function (2) f=0.25x+1.2

where x is the CHM tree height at the target pixel, and f gives the window radius.

The maximum search was thresholded to ignore maxima below five m, because the great majority of crowns were above this height and all ground points were below it. The linear window function form above was selected based on the recommendations of the package documentation, and the two parameters of the function were manually tuned, starting from their default values and ending with those presented above, until the segmentation results on the test data appeared to be appropriately segmenting the canopies to individual trees. This manual testing and tuning was performed iteratively across several input image plots with varying tree heights and degrees of openness to ensure that performance would be acceptable across the structurally heterogeneous woodland evaluated here.

Segmentation

Following identification of treetops, watershed segmentation was performed, again using the ForestTools package (Plowright, 2018). The SegmentCrowns() function was used with the treetops found above. This function performs a marker-controlled watershed segmentation, finding one segmented tree crown around each specified tree top marker from the function in the previous section. The function was used with the minimum canopy height parameter set to three m, preventing pixels below three m from being included in crowns. This means that while tree crown maxima were not permitted to occur below five m (in Section 2.2.4), tree crown edges were allowed to extend down to as low as three m. This height limit prevented inclusion of ground points while still retaining most of the canopy structure. The SegmentCrowns() function returns empty polygonal lines when the format parameter is set to “polygons,” and these were saved to a shapefile using the R function writeOGR().

Alignment

Input data

Alignment was performed between the provided ground and ITC datasets. The ITCs present in the dataset were divided up into training and test data as described in the parent paper (Marconi et al., 2018). The ground data consisted of stem IDs, locations in latitude and longitude, stem heights, and stem diameters. The remote sensing ITC data consisted of crown IDs, location in latitude and longitude, crown area, and plot IDs.

Algorithm

First, stems were divided into plots. This information was not provided in the input data, but plot IDs were included in the provided ITC crown data. Because the plots were small relative to the distance between plots, it was possible to cluster stem data into plots by simply iterating through the list and assigning each stem to the plot containing its nearest crown neighbor in latitude and longitude. In order to correct for the possibility of systematic error in ground-based estimation of location, all the stems in each plot were transformed by the difference in average coordinates between stem and crown groups within the plot.

Next, for each crown an estimate of crown diameter was formed using the known crown area and the assumption that crowns are roughly circular: (3) D=2(Aπ)0.5

Linear regressions were performed to find the relationships between the crown and stem diameter and between crown diameter and stem height. As well, multiple regression models with both of these predictive terms either with or without an interaction term were also run. Stem diameter was found to predict crown diameter more strongly than did stem height. As well, neither addition of stem height nor of both stem height and an interaction term between stem height and stem diameter was found to substantially improve the resulting correlation, so only the single regression model between stem diameter and crown diameter was used in further analysis. See the results section for greater detail on the statistical results of these regression models.

The RMS error between all crown diameter predictions based on this equation and all actual crown diameter values was determined across the training dataset. Analogously, the RMS error in latitude and longitude was found for all training crowns vs. the corresponding values for their paired stems.

Within each plot, a list of all possible pairings of stems and crowns was created and iterated through, with every pairing possibility being given a cost which was determined as follows: (4) C=(D−D^Drms)2+(X−X^Xrms)2+(Y−Y^Yrms)2

where C is the cost, D is crown diameter estimated from crown area, D^ is crown diameter predicted from the stem using the regression equation, Drms is the RMS error in diameter predictions, X is crown longitude, X^ is stem longitude, Xrms is the longitude prediction RMS error, Y is crown latitude, Y^ is stem latitude, and Yrms is the latitude prediction RMS error. The RMS error values were thus used to create dimensionless costs across the three variables, scaled by the overall noise in each signal. Within each plot, the set of pairings which minimized the total cost across all pairs was taken as the best alignment.

Classification

Input data

The input data for this task consisted of a dataframe of individual pixel observations within each tree. Each pixel was labeled with its associated crown ID and contained the height within the CHM at that point and the response in all the hyperspectral bands. Species labels were also provided for each crown ID within the training set. Again, see the parent paper (Marconi et al., 2018) for more information on the input dataset.

Algorithm

First, all the pixels in each tree were aggregated into a single observation of that tree, containing the average reflectance value in each band as well as the average height, the minimum height, and the maximum height across crown pixels, and the square root of the total number of pixels in the tree (as an approximation of crown diameter, since each pixel is 1 × 1 m).

The resulting tree vectors within the training set were grouped by species class. Two separate principle component analysis (PCA) routines were run on the four structural and 426 spectral parameters. The dimensionality of the set was reduced by retaining only the three most informative dimensions from the PCA result for the structural matrix and the 10 most informative dimensions from the spectral PCA result. The vectors within each class group were assembled into a prediction matrix for that class.

A set of maximum likelihood classifiers was built on the training data for each species class. Maximum likelihood methods work by assigning likelihoods L based on the following equation (Sisodia, Tiwari & Kumar, 2014): (5) L=(2π)−N/2|Y|−0.5exp(−0.5(t−T)T)Y−1(t−T)

where N is the total number of variables used in the classifier (here bands and structural parameters of trees), Y is the covariance matrix of the entire set of training vectors for the target class, t is the vector of parameters for the tree to be classified, and T is the average vector from the set of training vectors for the target class.

Each individual tree vector in the test set was transformed by the PCA parameters found above and used to determine a likelihood of assignment to each class. Trees were then assigned to the class with the highest likelihood score.

Results

Segmentation

The segmentation routine had the weakest results out of the three algorithms implemented here. The scoring used by the competition was based on the Jaccard index, which measures the overlap between two sets, and is given as follows: (6) J(A,B)=|A∩B||A∪B|=|A∩B||A|+|B|−|A∩B|

This index ranges between 0 and 1, with 0 representing a complete lack of overlap and 1 representing complete overlap. The index was calculated on the output trees compared to models of tree crowns produced by the competition coordinators hand-drawing crown segmentations on the data (Marconi et al., 2018). The algorithm used here yielded J = 0.184, compared to the baseline score found by the organizers of J = 0.0863 which used a simpler segmentation routine. Figure 1 below provides the Jaccard scores for all participants evaluated across tree sizes. The implementation in this study performed most strongly for larger trees, outperforming all other methods for the largest tree class, and outperformed the “baseline” method for trees of all sizes except for the 65 m2 area class.

Figure 1 Jaccard scores for each participating team, evaluated across crown diameters.

This figure was produced and provided by the competition organizers (Marconi et al., 2018).

An example segmentation of a plot scene is given in Fig. 2. Note the robustness of the routine to areas that do not contain tree crowns, with bare soil or vegetative ground cover visible.

Figure 2 Segmentation results for one plot.

(A) Raw RGB image of the plot scene. (B) Canopy height model of the scene, filtered to exclude points with NDVI <0.5 and height <3 m. Local maxima associated with presumptive treetops are shown in black marks. (C) Output of the watershed segmentation routine showing polygonal crowns. (D) Crown segmentations overlaid in red on the input RGB image. All RGB color imagery was created by NEON and is directly available within the dataset used in the competition (National Ecological Observatory Network, 2016). NEON’s data use policy allows for the free use and publication of all data contained within these datasets provided that credit is attributed.

Alignment

The regression equation found between crown diameter (as estimated by crown area) and stem diameter is given by (7) D^crown=0.114Dstem+0.882

where D^crown is crown diameter in m and Dstem is stem diameter in centimeter. This equation was found to be significant (p < 0.0001) with R2 = 0.697. A plot of crown diameters vs. stem diameters is given in Fig. 3. Also included in the figure is a plot of crown diameter vs. stem height, with regression equation (8) D^crown=0.127Hstem+2.152

where D^crown is crown diameter and Hstem is stem height, both in m. That model was also found to be significant (p < 0.01) but provided a relatively weak correlation with only R2 = 0.092, explaining relatively little of the variation in crown diameter.

Figure 3 (A) Linear regression for stem diameter (cm) vs. crown diameter (m) as estimated from crown area. (B) Linear regression for stem height (m) vs. crown diameter (m) as estimated from crown area.

Multiple regressions were also performed using both explanatory variables, one with an interaction term and one without. These respectively resulted in R2 values of 0.7106 and 0.7066, but while both models were significant overall, in both models all variables other than stem diameter failed to significantly predict crown diameter (p > 0.05). Because the stem diameter model outperformed the stem height model and addition of neither the stem height nor an interaction term in a multivariate model substantially improved the explanatory power of the model over stem diameter alone, only the simple linear regression between crown diameter and stem diameter was used for subsequent analysis.

As mentioned in the methods section, RMSE values in actual location (UTM easting and northing) vs. the location estimated from ITC segmentation were calculated, with respective values of 3.192 and 5.901. The RMSE value for the crown diameter predicted by the stem/crown diameter regression model vs. the actual crown diameter (approximated based on crown area) was also used, and was found to be 2.753.

The alignment routine performed better overall than segmentation, and was able to correctly align 48% of the input crowns to the associated trees. However, an identical performance was yielded by the benchmark routine used by the competition organizers. That routine was very similar to the one implemented here, but used only latitude and longitude to align trees and ignored stem and crown diameters.

Classification

The classification performance of algorithms in the competition was measured by two metrics—rank-1 score (recall, or the percentage of all test trees correctly classified) and cross-entropy score (which rewards participants for expressing uncertainty about predictions). Higher rank-1 scores and lower cross-entropy scores are associated with “better” classification results. The organizers also provided the precision and F1 scores of the competing submissions.

Classification yielded a rank-1 score of 0.8253, indicating that it correctly classified 82.53% of all the trees in the set, and had a cross-entropy score of 1.2247. This can be compared to the “baseline” method implemented by the competition organizers, which yielded a rank-1 score of 0.6667 and cross-entropy score of 1.1306. Figure 4 below provides a graph of precision, recall (or rank-1 accuracy), and F1 score results across the nine species classes used.

Figure 4 Classification results across tree species, presented in terms of precision (white), recall (gray), and F1 scores (black).

My algorithm was the only method other than the baseline which had a 100% success rate at correctly identifying the most common tree in the dataset (the longleaf pine, P. palustris). It also yielded a very high success rate at identifying the second most common species (the Turkey oak, Q. laevis), at 87.0%. However, it did not correctly identify any tree of any species other than these two most common species. Despite this, the two most common species represent a huge majority of the overall canopy in this system, covering 82.3% and 84.9% of all the trees in the training and test datasets, respectively.

Discussion

Segmentation

Segmentation remains a major challenge within the remote sensing community in systems with hardwood trees or substantial structural complexity and variability (Kaartinen et al., 2012; White et al., 2016; Heinzel & Koch, 2012), and individual tree segmentation algorithms are still often not fully deployed in commercial settings (Kaartinen et al., 2012; White et al., 2016). No group participating in this competition was able to yield a segmentation Jaccard score of higher than 0.34. This deficiency is likely exacerbated by the heterogeneity of canopy structure between test plots in this test system. Segmentation algorithms are often sensitive to canopy structure differences such as degree of openness (Naidoo et al., 2012; Kaartinen et al., 2012; White et al., 2016; Heinzel & Koch, 2012), and so in the future it might be beneficial to focus efforts on testing more algorithms which can automatically adjust their tuning based on the local openness and tree shape within different areas of the canopy. My approach used a single manual tuning of input parameters (although the window size used is variable), and work to automatically learn more effective parameters using training data represents a potential area for future improvement.

Alternatively, the linear regression model found to relate tree height to crown size, presented in 3.2, could simply be used for the window size model, eliminating these two parameters from the tuning effort. The NDVI filter parameter used can most likely be fixed at a constant value (such as the 0.2 used here) across all forestry applications, because all trees will be substantially higher in NDVI than are either the bare rock or water this parameter is meant to exclude. The height parameter should be varied across applications based on usage needs, especially the overall height of dominant trees in the target system and the question of whether understory trees and saplings in openings should be included. However, it is likely that any researcher or worker surveying a new system will already know this basic structural information, so the selection of these parameters need not be onerous.

Because most of the classification results submitted by teams in this competition were fairly powerful, it might be possible to perform pixel-wise classification to species first, and use this information to help inform the aggregation of those pixels into crowns (i.e., trees of different species cannot be part of the same crown). An alternative possibility is an iterative process which first segments and classifies crowns at the crown level and then splits crowns that appear to be combinations of two trees of differing species, or lumps adjacent crowns of the same species which might be different subcrowns within the same tree. Such solutions have been attempted before in other structurally heterogeneous systems, with moderate improvements reported over the original segmentation (Heinzel & Koch, 2012).

It may also be worthwhile to try pulling in more data for the segmentation routine than just CHM results, including hyperspectral information. As well, other segmentation algorithms could be considered which operate directly on the LiDAR point cloud, instead of the CHM. The CHM used here contained pixels 1.0 m wide, which reduces the height information available within a given tree by a factor of four relative to the previous analyses performed using 0.5 m pixels (Popescu & Wynne, 2004; Kaartinen et al., 2012) using the same segmentation technique. Point clouds typically contain more structural information than do the CHMs derived from them, and so direct operation on the point cloud could allow more useful height information to be gleaned from the data despite the dependence on coarser LiDAR scans than were used in other studies.

Alignment

I was surprised that the alignment algorithm did not perform more strongly, especially because the strongest alignment algorithm implemented by another team used a method very similar to the one implemented here (Dalponte, Frizzera & Gianelle, 2018). I believe that there may have been an error in the relative weighting of the shape and position terms, because the results here were identical to those produced by the “benchmark” algorithm which ignored crown area and used only tree positions. Further investigation will be warranted in future work.

However, the broader applicability of these alignment algorithms is limited, because they rely on one-to-one datasets of perfectly labeled crowns and stems. They cannot perform on datasets in which not every stem is labeled, which is likely the case in a real forestry application. Also, the pairing algorithm implemented here is suited to this competition but scales poorly with larger plot sizes—if it were to be implemented on a forest scale it would take an extremely long time to process all the potential pairings. Algorithms to automatically section the forest for alignment analysis into plots like those used here may be an area of potential interest for future work. Alternatively, it may be beneficial to rely instead on algorithms in which each target stem is compared to all of its nearest neighbor crowns, possibly with some cost to penalize multiple stems being mapped to the same crown.

Classification

The classification routine was extremely effective at identifying common species. In some contexts, this may be all that matters. For example, this would be entirely sufficient if the primary intent is to quantify the numbers in a community of a few very common “dominant” species. This would be the case for efforts to take inventory of a wood production forest, or to calculate parameters related to gross system function like primary productivity, water filtration, or carbon sequestration. In these cases, the small number of misidentified rare species may not be important.

In other cases, this kind of result would not be acceptable. If the goal is to identify rare species in a community so that they can be managed for conservation, ability to recognize uncommon species may be important. However, while this algorithm was aimed primarily at identifying the most common species, it should be noted that no algorithm was able to yield strong performance on all the uncommon species. In fact, the best other submission for uncommon species identification was still only able to recognize four of the seven uncommon classes with greater than a 50% success rate—and this came at a loss in accuracy of about 10% at identifying the second most common species. Further work is warranted to develop algorithms that are capable of robustly recognizing rare species based on very sparse training data.

Conclusion

This submission to the data science competition includes a tree segmentation, alignment, and classification pipeline which performs most strongly for common tree species. Consequently, it may be appropriate for applications such as maintenance of highly managed forestry plantations and efforts to estimate gross forest parameters in natural systems. Future work will focus primarily on improving the results of the segmentation algorithm, with emphasis also on improving the alignment of remotely collected and hand-labeled ground data. The latter will become especially important as the competition moves toward more realistic tree selection, potentially with overlapping plots and incomplete correspondence between the aerial and ground datasets.

As remote sensing methods continue to develop and the cost of deployment continues to decrease (with more and cheaper sensors and small aircraft), the technologies targeted by this competition may become increasingly important in a diverse array of disciplines, from agriculture to forestry to ecological research (Mulla, 2013). The newly introduced NIST DSE plant identification challenge may help to foster the development of systems for remote sensing analysis that are more streamlined and generalized across applications, which should aid their wider deployment across these fields. It is hoped that future competitions will continue to elaborate further on the methods developed here, and that this will aid in the expansion of remote sensing approaches into even more real-world applications and fields.

I thank the two anonymous reviewers and the editor for their extremely helpful and constructive feedback. Thanks are also due to the organizing team who structured the competition, provided data to the participants, and performed analysis of team submissions. Finally, I thank NEON for providing the data used herein.

Additional Information and Declarations

Competing Interests

Author Contributions

Data Availability

The authors declare that they have no competing interests.

Conor A. McMahon conceived and designed the experiments, performed the experiments, analyzed the data, contributed reagents/materials/analysis tools, prepared figures and/or tables, authored or reviewed drafts of the paper, approved the final draft.

The following information was supplied regarding data availability:

GitHub: https://github.com/conormcmahon/canopy_segmentation.

The data used in the project were produced by NEON. A link to NEON’s data use policy is below, which covers the freedom for users to publish the aerial RGB imagery included in Fig. 2.

http://data.neonscience.org/data-policy.

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
