# Peer review of "Remote sensing pipeline for tree segmentation and classification in a mixed softwood and hardwood system"

_PeerJ, doi:10.7717/peerj.5837_

## Round 0.1 · original submission · Minor Revisions

This paper is well-written and clear. The reviewers and I have only a few suggestions to improve the title and to fix a few acronyms in the text. Please respond to the reviewers suggestions.

Reviewer 1 ·

Basic reporting

The manuscript is relatively well-written and provides a nice context to the study. The manuscript can benefit from additional review on what has been done to establish such pipeline in the past (such as Kershaw et al., 2016; White et al., 2016; Hyyppä et al, 2012; van Leeuwen et al., 2010;).

Experimental design

The methods are well-described in most cases, except for omission of some minor details (please see general comments below).

It seems “maxima search” involved lots of visual evaluations before an appropriate set of parameters is arrived upon. Is there any way to present this process in more detail either in the methods or in the results section, so that readers know what exactly is involved in this process and this can be replicated?

Validity of the findings

The whole 'pipeline' required a lot of manual tweaking of the parameters, especially the NDVI/height thresholds and parameters used for local maxima. I am concerned that the pipeline presented here might have limited applicability beyond the current data and study area.

The results section can be improved. For example, in section 3.1, the author provided only an overall J score, but it gives no indication as to how the algorithm performed across trees of different sizes (crown, height) or species. The alignment was evaluated using RMSE and yet the RMSE values (overall or for different tree sizes/species) were not provided. The classification accuracies of different species were also not provided.

The discussion section is also weak. The author assumes that segmentation/ classification is major challenge in the remote sensing community because most participants did not perform well. Over the last two decades, the science of image segmentation/classification has matured enough, and I think the poor performance was solely because none of the participants used industry-standard approaches such as OBIA (e.g. Blaschke et al). The manuscript could benefit from additional comparison of the methods/results against previous tree segmentation/classification literatures (please refer to Zhen et al., 2016 or Blaschke, 2010 for overview)

Additional comments

- The title of the paper should highlight the core content of manuscript. It should be self-descriptive on its own, independent of other manuscripts submitted as part of the challenge.
- Lines 50-54: Are there appropriate citations of these data sets? If so, please cite them accordingly
- Line 55: State the full form of DSM, and that they are used to compute the canopy height model referred elsewhere in the manuscript
- Line 68. For clarity, please refer “the cloud” to something like “the point cloud derived from the raster pixels”
- Line 76: “landcover” to “land covered”
- Line 79: Mention that maxima search was done on NDVI-filtered LiDAR CHM. Also, briefly describe what the TreeTopFinder() does. (Popescu and Wynne, 2004?)
- Line 89: Briefly describe what segmentation algorithm SegmentCrowns() implements
- Lines 131-139: State how many structural and how many spectral parameters in total were used for PCA.
- Lines 160-162: Can the author also show how the relationship was when the crown height was also included?
- Line 203. Can the author elaborate what specific problem?

Reviewer 2 ·

Basic reporting

This is a well written manuscript. I enjoyed reading it. The author provided a good background to the problems investigated and used good illustrations to highlight the findings of the study. The data used in this study are publicly available (NEON and the data science competition).

Experimental design

The study has a clear design and the dataset and methods are explained in detail.

Validity of the findings

The segmentation method did not perform well, but the author acknowledged that this remains a challenge in the remote sensing field. The alignment and classification results are promising. The author’s conclusions are grounded in the results presented.

Additional comments

It was a pleasure to read the manuscript – good structure, clear explanations, and relevant background provided. I only have a small set of minor editing suggestions for the author.
Is the spelling LIDAR the most common one? I’ve seen mostly LiDAR and lidar in the literature.
Title: it might be good to make the title a bit more specific (for readers looking for papers on remote sensing and tree identification)
Abstract:
spell out the acronyms (NIST, DSE, NDVI, LIDAR, PCA); replace “compete in the competition” with “participate in the competition”
Background:
Line 20: use plural of aircraft
Line 38: spell out NIST DSE
Line 42: insert comma after “tasks”
Line 45: remove word repetition (“provided”)
Methods:
In 2.1: clarify which datasets are hyperspectral and which lidar; spell out DSM
Line 60: insert “(CHM)” after “canopy height model”
Line 68: spell out NDVI
Line 79: insert “R” before “package”
In Equation (2) is x the CHM value?
Line 81: reword “no ground points were”
Line 118: replace “predicated” with “predicted”
Discussion:
Line 202: Is (Dalponte and Gianelle) a citation? If yes, add year. If not, provide the name of the team mentioned here.
Conclusion:
Line 231: insert “data science” before “competition”
Figure 1 caption: spell out CHM

---

## Round 0.2 · accepted · Accept

Thank you for revising your paper and improving it in light of the reviews. I anticipate that this paper will be of interest not only to the remote sensing community but also to data-minded ecologists.

#